# Redox State Modulatory Activity and Cytotoxicity of *Olea europaea* L. (Oleaceae) Leaves Extract Enriched in Polyphenols Using Macroporous Resin

**DOI:** 10.3390/antiox13010073

**Published:** 2024-01-04

**Authors:** Tonia Luca, Giuseppe Antonio Malfa, Laura Siracusa, Alfonsina La Mantia, Simone Bianchi, Edoardo Napoli, Stefano Puleo, Angelo Sergi, Rosaria Acquaviva, Sergio Castorina

**Affiliations:** 1Department of Medical, Surgical Sciences and Advanced Technology, University of Catania, Via Santa Sofia, 95123 Catania, Italy; tluca@unict.it (T.L.); sergio.castorina@unict.it (S.C.); 2Department of Drug and Health Sciences, University of Catania, Viale A. Doria 6, 95125 Catania, Italy; alfy.lamantia@gmail.com (A.L.M.); simone.bianchi@phd.unict.it (S.B.); angelo.sergi@rocketmail.com (A.S.); racquavi@unict.it (R.A.); 3Research Centre on Nutraceuticals and Health Products (CERNUT), University of Catania, Viale A. Doria 6, 95125 Catania, Italy; 4Institute of Biomolecular Chemistry, Italian National Research Council ICB-CNR, Via Paolo Gaifami 18, 95126 Catania, Italy; laura.siracusa@icb.cnr.it (L.S.); edoardo.napoli@icb.cnr.it (E.N.); 5Mediterranean Foundation “GB Morgagni”, 95125 Catania, Italy; spuleo@unict.it

**Keywords:** phytochemicals, polyphenols, plant extract, necrosis, ROS, oxidative stress, glutathione, prostate cancer

## Abstract

The food products derived from *Olea europaea* are a fundamental part of the Mediterranean diet, and their health-promoting effects are well known. In this study, we analyzed the phytochemical characteristics, the redox state modulatory activity, and the cytotoxic effect of an olive leaf aqueous extract enriched by macroporous resin on different tumor and normal cell lines (LNCaP, PC3, HFF-1). HPLC-DAD analysis, the Folin–Ciocalteu and aluminum chloride methods confirmed the qualitatively and quantitatively high content of phenolic compounds (130.02 ± 2.3 mg GAE/g extract), and a DPPH assay (IC_50_ = 100.00 ± 1.8 μg/mL), the related antioxidant activity. The biological investigation showed a significant cytotoxic effect, highlighted by an MTT test and the evident cellular morphological changes, on two prostate cancer cell lines. Remarkably, the extract was practically non-toxic on HFF-1 at the concentrations (100, 150, 300 µg/mL) and exposure times tested. Hence, the results are selective for tumor cells. The underlying cytotoxicity was associated with the decrease in ROS production (55% PC3, 42% LNCaP) and the increase in RSH levels (>50% PC3) and an LDH release assay (50% PC3, 40% LNCaP, established necrosis as the main cell death mechanism.

## 1. Introduction

Many components of the Mediterranean diet are strictly correlated with a lower risk of various types of cancer [1,2]. In particular, the protective role of olive oil on digestive, breast, and prostate cancer has been highlighted by several findings [3]. These beneficial effects are mostly due to the secondary metabolites present in the drupe and leaves of *Olea europaea* L. (Oleaceae) [4,5]. In the traditional medicine of many Mediterranean countries, remedies from olive leaves have been largely used for the treatment of various diseases, including cancer [6]. Research data also corroborate the health properties of olive polyphenols due to their antioxidant, anti-inflammatory, and anti-cancer activities [7]. In previous in vitro studies, olive polyphenols such as hydroxytyrosol and oleuropein have demonstrated the ability to inhibit the proliferation of different cancerous cells, including prostate tumor cell lines [8].

Prostate cancer is the most commonly diagnosed malignancy in 112 countries and the most frequent cause of death in 48 countries, representing a leading cause of male death all over the world [9]. Its etiology has highlighted a positive family history and advancing age as well-established risk factors, together with black race/ethnicity [10]. Black men in the United States and the Caribbean, in fact, are the most affected [11]. Some genetic mutations (e.g., BRCA1 and BRCA2) and conditions (Lynch syndrome) can be also considered [12]. The risk of advanced prostate cancer is increased by environmental and lifestyle factors, including obesity [13], diabetes mellitus [14], nutritional factors, and smoking [15].

At the end of the 1980s and the beginning of the 1990s, prostate-specific antigen (PSA) testing was introduced, and this allowed the detection of preclinical cancers, leading to an increase in incidence rates in the United States, Canada, and Australia [10]. On the contrary, declines in the incidence in the late 2000s could be due to a reduction in the use of PSA testing because of modifications in the guidelines regarding PSA-based screening of asymptomatic men [16,17]. Similar patterns were observed in northern and western Europe, but they were less marked because of a later and slower adoption of PSA testing [18].

Treatment of prostate cancer depends on the risk category of the disease. Active surveillance is a choice for low-risk and intermediate-risk disease patients [19]. Chemotherapy can be considered for patients with recurrent or metastatic, castrate-sensitive, or resistant disease [20,21]. Because androgens and androgen deprivation have deep effects on the immune system, immunotherapy has recently been considered among new treatment options in association with androgen deprivation therapy to improve survival for patients with advanced, metastatic prostate cancer [22].

Some new approaches aim to reduce disease progression or recurrence in patients with known disease. In these cases, risk factors such as lifestyle, physical activity, and nutrition are monitored and, if necessary, modified. In regard to nutrition, micro-, and macronutrients (Vitamins D and E, selenium, tomato, and lycopene) and herbal supplements (soy, green tea, and saw palmetto berry extract), for example, have been proposed to be useful for the prevention of prostate cancer or as substances to help prostate health [23].

However, dietary and nutrition may not be effective, and because most treatments of recurrent and/or metastatic prostate cancer pose the possibility of serious side effects, it is particularly important to develop effective and low-toxicity anti-prostate cancer drugs.

In this study, a polyphenol-enriched extract from olive leaves (OLEE) was tested to evaluate its potential anticancer properties on two prostate cancer cell lines, LNCaP and PC3. For this purpose, the extract was analyzed spectrophotometrically for the quantitative determinations of total polyphenols and total flavonoid content and by HPLC-DAD/HPLC-ESI-MS and a 2,2-diphenyl-1-picrylhydrazyl (DPPH) assay for qualitative and antioxidant characterization, respectively. A cell proliferation assay was performed, and morphological analysis, Lactate dehydrogenase (LDH) release, and antioxidant activities (reactive oxygen species and total thiol groups) in both cell lines were evaluated.

## 2. Materials and Methods

### 2.1. Chemicals and Reagents

2,2-diphenyl-1-picrylhydrazyl (DPPH), 2′,7′-dichlorofluorescein diacetate (DCFH-DA), 5,5-ditiobis-2-nitrobenzoic acid (DTNB), dimethyl sulfoxide (DMSO), and analytic-grade organic solvents were purchased from VWR (Milan, Italy). Unless otherwise specified, all other chemicals were brought from Sigma–Aldrich (Milan, Italy).

All materials and media for cell culture were bought from ThermoFisher Scientific (Monza, Italy) unless otherwise specified.

### 2.2. Preparation of Olive Leaf Enriched Extract (OLEE)

The leaves of *Olea europea* L. from different cultivars (*Tonda Iblea*, *Biancolilla*, *Carolea*) were collected randomly in an olive grove on the southern seacoast of Syracuse (Sicily, Italy) at the end of April 2021. The plant matrix was authenticated by the pharmaceutical botanist G. A. Malfa. A voucher specimen of the plants (No. 04/21) was deposited in the herbarium of the Department of Drug and Health Sciences, Section of Biochemistry. Freshly washed and wiped leaves were dried at 50 °C in a ventilated oven for 48 h and subsequently crushed into small particles. The crude extract was obtained by infusion in distilled water (1:8 = matrix weight: water volume) at 70 °C with continuous stirring for 2 h. The extraction procedure was repeated twice.

The supernatants were gathered and hot filtered with Whatman n° 4 filter paper. For the total polyphenols enrichment of the extract, macroporous resin column chromatography was performed in a glass column of 20 cm height and an inner diameter of 2.3 cm filled with 12 g of amberlite polymeric resins FPX-66 (DuPont, Wilmington, DE, USA), activated according to the manufacturer’s instructions. The olive leaf crude extract was put into the column at a rate of 1 BV/h until saturation was reached. In order to remove the residual feed, the saturated resin was washed with cold distilled water (4 °C) at a rate of 2 BV/h for 1.5 h and then drained. Polyphenols were desorpted from the resins with absolute ethanol for 24 h at room temperature. The extraction procedure was repeated three times. The gathered alcoholic solution was brought to dryness under reduced pressure with a rotatory evaporator. The polyphenol yield on the vegetal matrix was 0.265% gallic acid equivalents (GAE), and the enriched extract yield was 14.9 g/100 mL of resin.

### 2.3. Phytochemical Analysis

#### 2.3.1. Spectrophotometric Determinations of Total Polyphenols and Total Flavonoids

The total phenolic concentration of OLEE was measured spectrophotometrically (λ = 750 nm) by the Folin–Ciocalteu method, as reported by Acquaviva et al. [24]. Gallic acid (GA) was used as a standard, and the results were expressed as milligrams of gallic acid equivalents (GAE)/g extract (dw) ± S.D. The total flavonoid content of OLEE was measured by the aluminum chloride method. Catechin (C) was used as a standard, and the results were reported in milligrams of catechin equivalents (CE)/g extract [24]. Each result represents the mean ± S.D. of three experimental determinations.

#### 2.3.2. HPLC-DAD and HPLC-ESI-MS Analyses

A small aliquot (ca. 10 mg) of OLEE was solubilized in 70% ethanol, opportunely diluted, filtered, and sent to analytical determinations.

High-performance liquid-chromatographic analyses were carried out on an Ultimate3000 instrument equipped with a binary high-pressure pump, a Photodiode Array detector, a Thermostated Column Compartment, and an Automated Sample Injector (Thermo Scientific, Milan, Italy). Collected data were processed through a Chromeleon Chromatography Information Management System v. 6.80. Chromatographic runs were performed using a reverse-phase column (Gemini C18, 250 × 4.6 mm, 5 μm particle size; Phenomenex, Torrance, CA, USA) equipped with a guard column (Gemini C18 4 × 3.0 mm, 5 μm particle size; Phenomenex). The extract was analyzed according to Gambacorta et al. [25]. The diode array detector (DAD) was set in the range from 600 to 190 nm, recording the chromatograms at 280, 330, and 350 nm. HPLC-ESI-MS analyses were performed using the same conditions (solvents, elution program, guard column, column, injection volume, and flow) described above, while the ESI mass spectra were acquired using an Exactive Plus Orbitrap MS (ThermoFisher Scientific, Inc., Milan, Italy) and a heated electrospray ionization interface. Mass spectra were recorded while operating in the negative ion mode, in the 120–1500 *m*/*z* range at a resolving power of 25,000 (full-width-at-half-maximum at *m*/*z* 200). This resulted in a scan rate of >1.5 scans/s when using the automatic gain control target of 1.0–106 and a C-trap injection time of 250 ms. This was performed under the following conditions: capillary temperature 300 °C nebulizer gas (nitrogen) with a flow rate of 60 arbitrary units, auxiliary gas flow rate of 10 arbitrary units, source voltage 3 kV, capillary voltage 82.5 V, and a tube lens voltage 85 V. The Orbitrap MS system was tuned and calibrated in the positive mode, by the inclusion of standard solutions of sodium dodecyl sulphate (Mr 265.17 Da), sodium taurocholate (Mr 514.42 Da), and Ultramark (Mr 1621.00 Da). Data acquisition and analyses were performed using the Excalibur software version 4.3 [26]. Analyses were always carried out in triplicate.

### 2.4. Antioxidant-Free Cell Assay

#### DPPH Test

The free radical-scavenging capacity of the extract (100–200–400 μg/mL) was performed spectrophotometrically (λ = 517 nm) using DPPH and compared to Trolox (30 µM) as a reference compound [27]. Results were obtained from the mean of three independent experiments and reported as the mean 50% inhibitory concentration (IC_50_) ± S.D.

### 2.5. Cell Lines

The human-derived metastatic prostate carcinoma cell line LNCaP was obtained from American Type Culture Collection (CRL-1740, ATCC^®^, Manassas, VA, USA). The human prostate cancer cell line PC3 was provided by the Department of Biomedical and Biotechnological Sciences, University of Catania, Italy. LNCaP cells were maintained in Roswell Park Memorial Institute (RPMI)-1640 medium with a GlutaMAX™ supplement and supplemented with fetal bovine serum 10% (*v*/*v*) (FBS) and antibiotics (100 units/mL penicillin, and 100 μg/mL streptomycin).

Routine maintenance of PC3 cells was in Dulbecco’s Modified Eagle’s Medium (DMEM) medium supplemented with 10% (*v*/*v*) FBS and antibiotics (100 units/mL penicillin, and 100 μg/mL streptomycin).

Human foreskin fibroblasts (HFF-1), obtained from the American Type Culture Collection (SCRC-1041, ATCC^®^, Rockville, MD, USA), were cultured in (DMEM) supplemented with 15% (FBS), 4.5 g/L glucose, 100 units/mL penicillin, and 100 μg/mL streptomycin. All the cells were cultured in a 5% CO_2_-equilibrated 37 °C incubator.

#### 2.5.1. Cell Viability Assay

HFF-1, LNCaP, and PC3 cells were plated in complete medium containing 10% (*v*/*v*) FBS at 1.8 × 10^3^ cells/well in 96-well plates. After 24 h, the medium was replaced with complete medium, supplemented with 1% (*v*/*v*) FBS. After a further 24 h, cells were treated for 48 h and 72 h with 1% (*v*/*v*) FBS medium containing OLEE previously solubilized in DMSO at concentrations of 100 µg/mL, 150 µg/mL, and 300 µg/mL. Control cells received medium containing 1% (*v*/*v*) FBS or DMSO at the higher concentration present in the treatment (0.6%) in medium containing 1% (*v*/*v*) FBS. The experiment was performed in triplicate with six technical replicates.

At the end of the treatments, 3-(4,5 dimethylthiazol-2-yl)-2,5-diphenyl tetrazolium bromide (MTT) solution (5 mg/mL in phosphate-buffered saline) was added to the medium, as previously described [28]. Briefly, after treating with OLEE, 10 μL 5 mg/mL MTT solution was added to each well, and the plates were incubated for 3 h at 37 °C. The culture medium was then removed, and precipitated formazan crystals were solubilized in DMSO (200 μL). Absorbance data of each well were read at 550 nm, normalized to a percentage of vehicle-treated control, and graphed. Results were expressed as a percentage of cell viability vs. untreated control cells.

#### 2.5.2. Light Microscopy and Morphological Analysis

PC-3 and LNCaP cells were plated in a complete medium containing 10% (*v*/*v*) FBS at a density of 20 × 10^3^ cells/well in 24-well plates. After 24 h, the medium was replaced with a complete medium supplemented with 1% (*v*/*v*) FBS. After a further 24 h, cells were treated with 100 µg/mL and 150 µg/mL OLEE for 24 h and 48 h. Following treatment, cell viewing and analysis of the number and shape were performed using an image analysis system consisting of an inverted microscope, a digital camera, and an imaging workstation personal computer. Axio-Vision Release 4.8.2-SP2 Software (Carl Zeiss Microscopy GmbH, Jena, Germany) was used for the acquisition and analysis of images.

#### 2.5.3. LDH Release

LDH activity, assessed by determining the β-nicotinamide-adenine dinucleotide (NADH) absorbance reduction, was measured in culture medium and cell lysates separately at λ= 340 nm [29]. Briefly, cells were seeded in a 6-multiwell plate (4 × 10^4^ cells/well), and after 24 h, the medium was replaced with complete medium supplemented with 1% (*v*/*v*) FBS. After a further 24 h, cells were treated with the different concentrations of extract (100–150 µg/mL) for 48 h. The increased LDH activity in the culture medium showed a relationship with the percentage of dead cells. Results were expressed as a percentage of LDH released and are the mean ± S.D. of five independent experiments in triplicate.

### 2.6. Intracellular Redox State Evaluation

#### 2.6.1. ROS Determination

LNCaP and PC3 cells were plated in a 6-multiwell plate (4 × 10^4^ cells/well), and after 24 h, the medium was replaced with complete medium supplemented with 1% (*v*/*v*) FBS. After a further 24 h, they were treated with 100 μg/mL and 150 μg/mL OLEE for 48 h, and Reactive Oxygen Species (ROS) levels were determined by using a fluorescent probe, DCFH-DA [27]. The fluorescence of the oxidized radical species 2′,7′-dichlorofluorescein (DCF) was recorded spectrofluorometrically (excitation, λ = 488 nm; emission, λ = 525 nm). The protein content was determined using the Sinergy HTBiotech instrument by measuring the absorbance difference at λ = 280 and λ = 260. The results were reported as a percentage of fluorescence intensity/mg protein with respect to control (untreated). Values are the mean ± S.D. of five independent experiments in triplicate.

#### 2.6.2. RSH Determination

The levels of non-protein thiol groups (RSH) were measured using an assay based on the reaction of thiol groups with DTNB to give a colored compound absorbing at λ = 412 nm [30]. The protein content was determined using the Sinergy HTBiotech instrument by measuring the absorbance difference at λ = 280 and λ = 260.

Results are expressed as the percentage in nanomoles of RSH/mg of protein with respect to control (untreated) cells and represent the average ± S.D. of five independent experiments in triplicate.

### 2.7. Statistical Analysis

The experimental data are presented as the mean ± standard deviation (S.D.). We performed statistical analysis of the data using one-way analysis of variance (ANOVA) followed by Tukey’s post hoc test in Graph Prism version 5. Differences were considered significant when *p* ≤ 0.05.

## 3. Results

### 3.1. Phytochemical Analysis

#### 3.1.1. Determination of Polyphenolic Profile

The polyphenols and flavonoids content of OLEE was 130.02 ± 2.6 mg of gallic acid equivalent/g (mg GAE/g extract) and 70.13 ± 1.2 mg of catechin equivalent/g extract (mg CE/g extract), respectively (Table 1, columns 1 and 2). These data were confirmed by HPLC-DAD analysis.

#### 3.1.2. Compositional Analyses on OLEE

A series of spectroscopic and spectrometric analyses were carried out on the OLEE object of this study, thus revealing a very complex matrix with more than 30 different chromatographic signals almost entirely belonging to the category of polyphenols (Figure 1). By cross-referencing data coming from UV-Vis and mass spectra, several subclasses of phenolic compounds were identified, namely hydroxytyrosol and derivatives (peaks 1, 3 and 4), flavanones (peaks 6, 8, 15 and 16) and flavones (luteolin, apigenin), together with their glycosylated derivatives, especially rutinosides (rutinose = glucose + rhamnose). All the main peaks have been identified, as shown in Table 2. A peculiarity of this extract is the presence of many polymethoxy derivative compounds (peaks 23–27), as further evidenced by the redshifts in their UV-Vis spectra and their peculiar mass traces.

### 3.2. In Vitro Cell-Free Antioxidant Properties

#### DPPH Assay

In the DPPH assay used to test the quenching effect, the extract showed significant antioxidant properties and an IC_50_ value of 100 ± 3.11 µg/mL equivalent to 15 µM ± 0.62 Trolox (Table 1, column 3).

### 3.3. Cytotoxicity on Normal and Cancer Cells

#### 3.3.1. Effect of OLEE on Cell Viability and Morphology

Prostate cancer cells were treated with different concentrations of OLEE for 48 h and 72 h. We observed significantly decreased cell proliferation of PC-3 and LNCaP cells after 48 h of exposure to 150 µg/mL and to 100 µg/mL for LNCaP cells only. After 72 h of treatment, the cytotoxic effect was increased. In HFF-1, used as non-tumor control cells, the extract did not affect cell viability at any concentration tested (Figure 2). Since cell viability was already significantly affected at 48 h, this exposure time was chosen as the longest duration of treatment for the subsequent investigations.

In the following experiment, prostate cancer cells were treated with 100 µg/mL and 150 µg/mL OLEE for 24 h and 48 h and examined by inverted light microscopy. Morphological changes were already observed after 24 h treatment. They were particularly evident after 48 h, while cells treated only with vehicle remained normal in size and shape (Figure 3).

#### 3.3.2. LDH Release

In order to confirm the cytotoxic effect of the extract and to investigate cell death, we performed an LDH assay. The results, expressed as the percentage of LDH release, clearly confirmed that OLEE at different concentrations (100–150 µg/mL) reduced the cell viability of LNCaP and PC3 cells by inducing necrotic cell death. Figure 4 shows that at the highest concentration (150 µg/mL), about 50% and 40% LDH release for PC3 and LNCaP cells, respectively, was reached.

### 3.4. Intracellular Redox State Evaluation

#### 3.4.1. Determination of ROS Levels

The possible modulatory activity of the cellular redox state was evaluated by the determination of ROS levels in the cells. Treatments of PC3 and LNCaP cells with 100 µg/mL and 150 µg/mL OLEE were able to reduce ROS levels in a dose-dependent manner. At 150 µg/mL OLEE, the ROS content was reduced by about 55% and 42% in PC3 and LNCaP cells, with respect to untreated cells (Figure 5).

#### 3.4.2. Determination of RSH Content

Non-proteic thiol groups are mainly represented by glutathione, the major endogenous nonenzymatic antioxidant. The amount of glutathione in cells is strictly linked to the cellular redox homeostasis. Analysis of the content of non-proteic thiol groups in PC3 and LNCaP cells, treated with different concentrations of the extract (100–150 µg/mL), showed a significant increase in RSH groups (Figure 6). In PC3 cells, the extract, at all concentrations, increased RSH levels compared to untreated cells, while in LNCaP cells, only the highest concentration increased RSH levels slightly with respect to untreated cells.

## 4. Discussion

It is now almost a certainty that food and natural substances play a fundamental role in reducing the incidence of various types of pathologies [31]. They have also been shown to be able to prevent multiple forms of cancer or to act as anticancer agents in combination with other drugs [32]. Different compounds can be easily obtained from several natural products, such as fruits, vegetables, seeds, cacao, coffee, tea, oil, wine, and beer, and for most of them, nutraceutical properties are known [33].

In Mediterranean countries, the use of olive products, including leaves, for the treatment of various disorders is well documented, and the health effects reported are certainly related to the secondary metabolites found in the olive vegetal matrix (leaf and fruit) [34]. Several studies have shown that these compounds are able to prevent and/or reduce the progression of cardiovascular, neurodegenerative, and tumor diseases [3]. In this study, the spectrophotometric quantitative characterization of OLEE revealed a good amount of phenolic compounds (130 mg GAE/g extract), of which about 54% was represented by flavonoids. Previous studies adopting different extraction methods reported lower values in total polyphenols and flavonoid content with regard to those shown in this work [35,36], confirming the efficacy of the enrichment process by absorption on macroporous resin. By HPLC/DAD and HPLC-ESI-MS analysis, 27 peaks (Figure 1) were identified (Table 2); some of the main compounds were hydroxytyrosol, luteolin, apigenin derivatives, and others such as methoxylated polyphenol derivatives, characterized by the presence of methoxy groups (-OCH_3_) attached to the aromatic rings; compounds, which have already been identified in this plant matrix [37]. Although methoxylated polyphenols exhibit less antioxidant activity, they are generally more lipophilic than non-methylated ones [38], with the capacity to cross the cell membrane faster and accumulate in the cytoplasm [39]. Despite the presence of methoxylated polyphenol derivatives, the DPPH assay extract showed a good radical scavenging activity for OLEE with an IC_50_ of 100 ± 1.8 μg/mL. Despite different results for the antioxidant activity of olive leaf extracts, the comparison with these is difficult due to the various experimental conditions, methods, and plant matrix used. Mansour et al. [40] found an IC_50_ of approximately 50 µg/mL in aqueous olive leaf non-enriched extracts. This higher antioxidant activity is probably related to the different phytochemical profiles.

Starting from these results, we tested the cytotoxicity of OLEE at different concentrations and exposure times on two prostatic cancer cell lines, PC3 and LNCaP, and in healthy human fibroblasts HFF1. The enriched extract showed remarkable selectivity against the two cancer cell lines, evidencing cytotoxic effects with a starting concentration of 100 μg/mL at 24 h of treatment for LNCaP cells and 72 h for PC3. Conversely, no significant modifications were recorded for healthy HFF1 cells. Earlier, the cytotoxicity of olive leaf extracts was studied on different healthy and cancer cell lines, including prostate cancer cell lines, and reported the cytotoxic selectivity on tumor cells compared to healthy fibroblasts, which sowed little sensitivity to the phytochemicals of olive leaf [41,42]. The cytotoxic activity of OLEE in this experimental model was evidenced by the increment in LDH release (Figure 4) in both LNCaP and PC3 cells, indicating the induction of membrane breakdown and consequent cell death. Previous studies showed that several plant extracts can induce a specific necrotic cell death called necroptosis in different tumor cell lines [43,44] by inducing metabolic stress [45]. In our experimental model, a metabolic stress condition can be supposed by the reduced mitochondrial activity recorded in the MTT assay. Considering the promising antioxidant activity highlighted by the DPPH test, we investigated the ROS production and RSH content in PC3 and LNCaP cells treated with OLEE at cytotoxic concentrations. It has been demonstrated that oxidative stress plays a substantial role in prostate cancer, contributing to the processes of carcinogenesis, progression, and invasiveness, including the onset of the androgen-independent phenotype [46]. Moreover, free radicals, such as ROS, are implicated in prostate cancer cell growth and survival [47,48].

Particularly, PC3 and LNCaP cells are well known to exhibit higher levels of ROS that are proportionally linked to aggressive phenotype [49], and their modulation is responsible for affecting cell growth [50]. On both cell lines, the enriched extract markedly reduced ROS production, decreasing the levels by about 50% at the concentration of 150 μg/mL with respect to the untreated cells. This remarkable effect on intracellular redox homeostasis can be linked to the direct antioxidant activity of the extract, demonstrated by the DPPH test results (IC_50_ of 100 ± 1.8 μg/mL). A precedent study showed that the inhibition of ROS generation and partial neutralization selectively block the growth and proliferation of prostate cancer cells [49]. The antiproliferative activities of OLEE were further confirmed by the results on RSH amount in treated cancer cells, where at the concentration of 150 μg/mL, the phytocomplex contained in the extract significantly increased RSH levels in LNCaP and firmly in PC3 cells. Glutathione levels, a biomarker closely related to the RSH amount, is reported to be increased during the non-proliferative cellular phase both in PC3 and LNCaP cells [48]. All the above data suggest the probability that OLEE exerted its antiproliferative and cytotoxic activities through the induction of cellular metabolic distress mediated by the destabilization of redox homeostasis in prostate cancer cells both directly by its radical scavenger activities and indirectly, possibly by the inhibition of ROS production and an increased glutathione amount. The suffering cellular state induced in PC3 and LNCaP cells was confirmed by morphological changes and the necrotic cell death displayed in treated cells. These activities are probably related to the particular phytochemical profile of the extract, which is rich in methoxylated polyphenol derivatives, more capable of crossing the cell membrane, and able to induce selective cytotoxicity in tumor cells significantly. Furthermore, the ability to modulate the dysregulated redox state of tumor cells is gaining importance in cancer therapy [51]. Additional studies are ongoing to deeply investigate the possible molecular pathways modulated by the phytocomplex responsible for the anticancer activity in androgen independent and dependent prostate cancer cells. 

## 5. Conclusions

A key node for sustainable development is the valorization of agri-food byproducts as a cheap and sustainable source of phytochemicals for developing natural ingredients for well-being industries, even using green technologies. The results reported in this study suggest that the olive leaves’ aqueous enriched extract could be a good and promising natural ingredient with cellular protective properties due to its selective cytotoxicity toward prostate cancer cells and its ability to modulate cellular redox homeostasis. However, additional studies are needed before its possible application as a nutraceutical or phytotherapeutic ingredient.

## Figures and Tables

**Figure 1 antioxidants-13-00073-f001:**
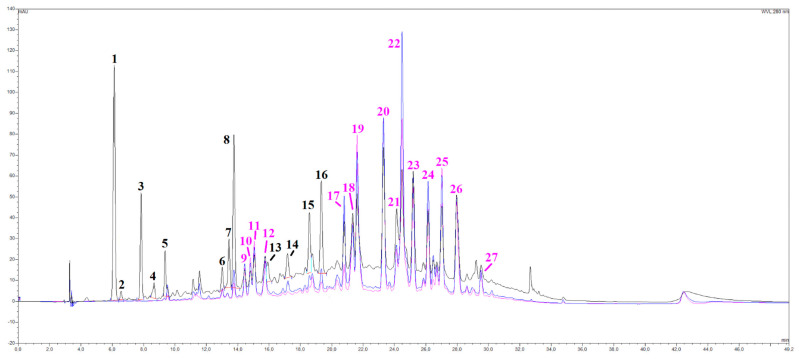
HPLC/DAD chromatograms, visualized at 280 nm (black line), 330 nm (blue line), and 350 nm (pink line) of the OLEE. See the text for further information and experimental details.

**Figure 2 antioxidants-13-00073-f002:**
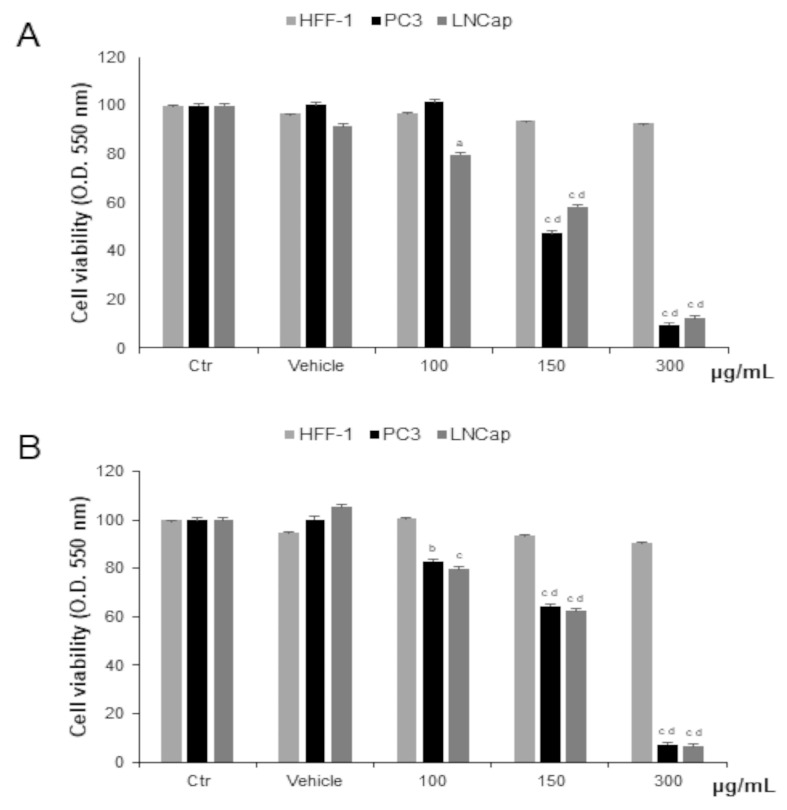
Cell viability in HFF-1, LNCaP, and PC3 cells untreated (Ctrl) and treated with vehicle or with OLEE for 48 h (**A**) and 72 h (**B**). Values are the mean ± s.e.m. of three different experiments with six wells assigned to each treatment. ^a^ Significant vs. untreated control cells: *p* < 0.05; ^b^ Significant vs. untreated control cells: *p* < 0.001, ^c^ Significant vs. untreated control cells: *p* < 0.0001; ^d^ Significant vs. treated cells: *p* < 0.05.

**Figure 3 antioxidants-13-00073-f003:**
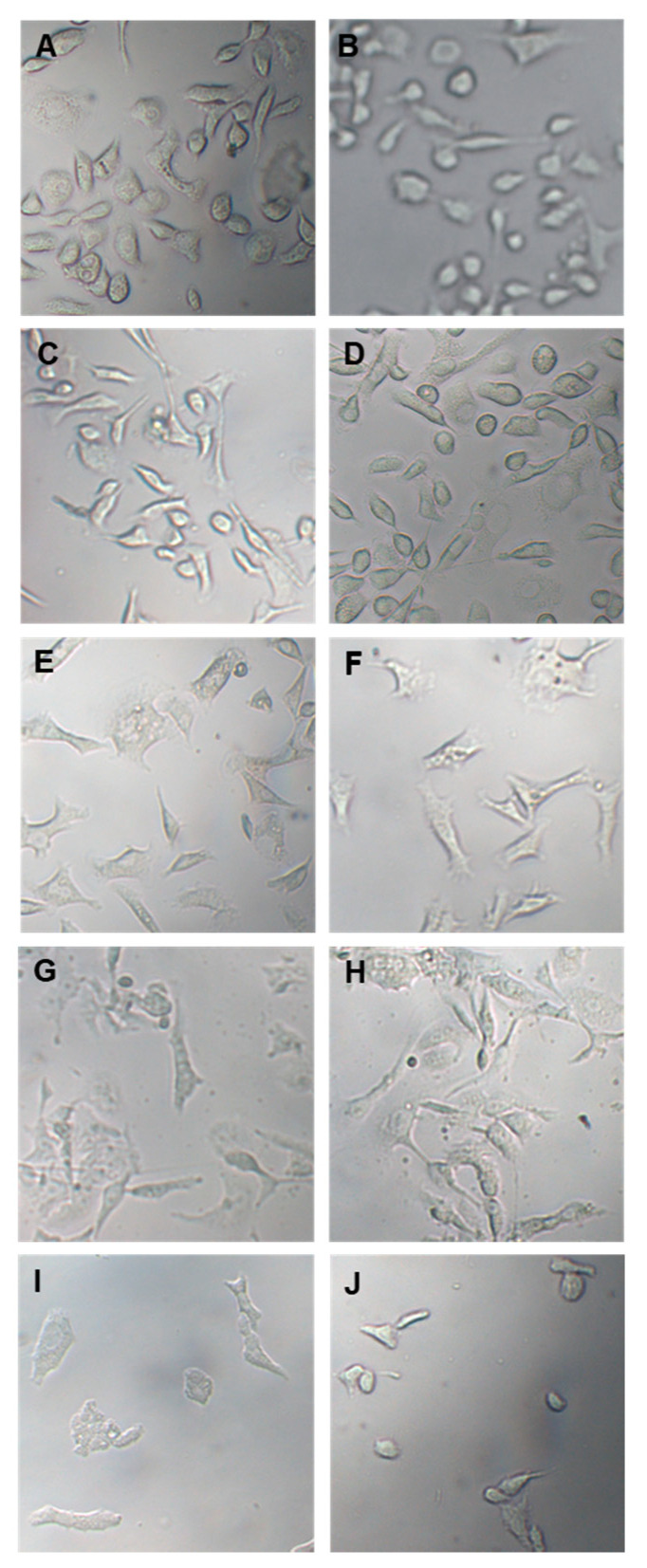
Optical microscope images of PC3 and LNCaP cells treated with vehicle or with OLEE for 24 h and 48 h. On the (**left**), PC-3 cells; on the (**right**), LNCaP cells. (**A**,**B**): Vehicle; (**C**,**D**): 100 µg/mL OLEE 24 h; (**E**,**F**): 150 µg/mL OLEE 24 h; (**G**,**H**): 100 µg/mL OLEE 48 h; (**I**,**J**): 150 µg/mL OLEE 48 h. Magnification 10× was used.

**Figure 4 antioxidants-13-00073-f004:**
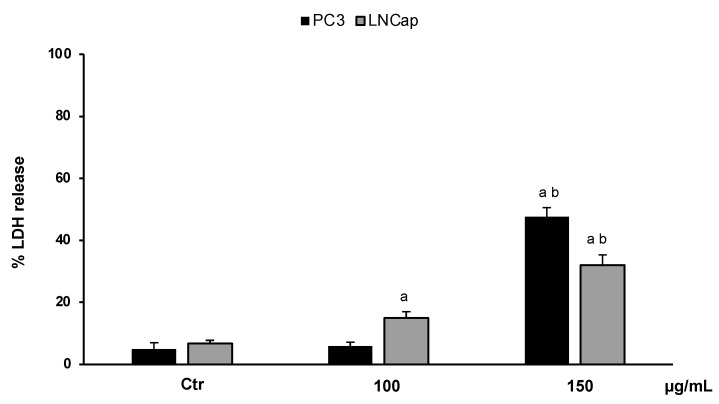
LDH release in untreated PC3 and LNCaP cells (Ctrl) and treated for 48 h with the extract (100–150 μg/mL). Values are the mean ± S.D. of five experiments in triplicate. ^a^ Significant vs. untreated control cells: *p* < 0.05; ^b^ Significant vs. treated cells: *p* < 0.05.

**Figure 5 antioxidants-13-00073-f005:**
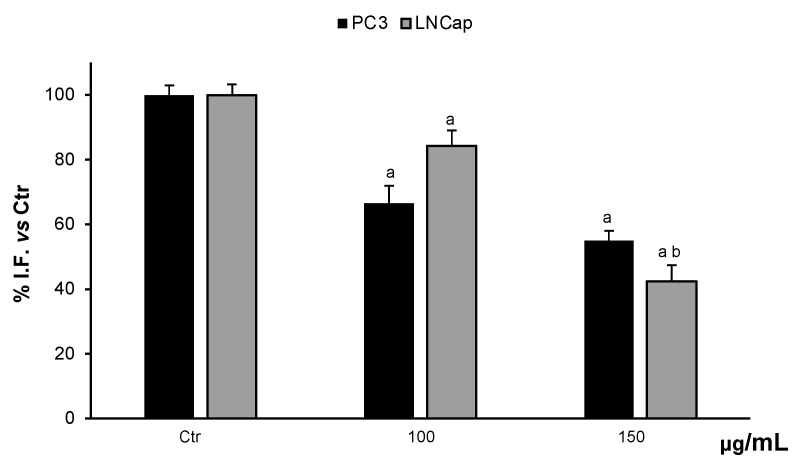
ROS levels in PC3 and LNCaP untreated cells (Ctrl) and treated for 48 h with extract (100–150 μg/mL). Values are the mean ± S.D. of five experiments in triplicate. ^a^ Significant vs. untreated control cells: *p* < 0.05; ^b^ Significant vs. treated cells: *p* < 0.05.

**Figure 6 antioxidants-13-00073-f006:**
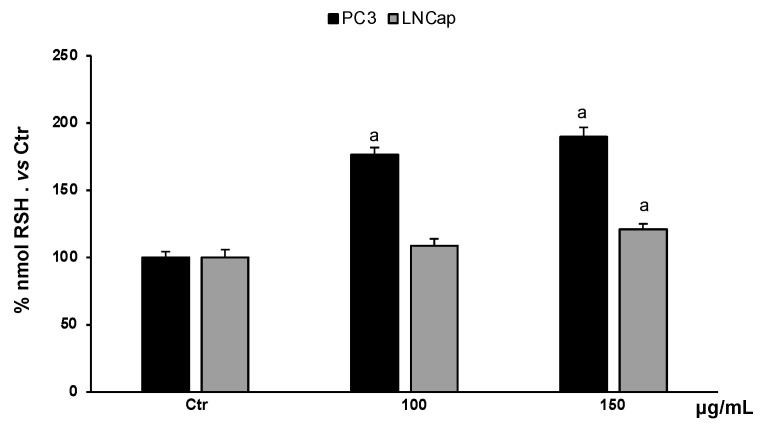
RSH levels in PC3 and LNCaP untreated cells (Ctrl) and treated for 48 h with extract (100–150 μg/mL). Values are the mean ± S.D. of five experiments in triplicate. ^a^ Significant vs. untreated control cells: *p* < 0.05.

**Table 1 antioxidants-13-00073-t001:** Spectrophotometric quantitative determination of total polyphenols, flavonoids, and DPPH test in OLEE extract.

	Total Polyphenols (mg GAE/g Extract)	Total Flavonoids (mg CE/g Extract)	DPPH Test IC_50_ (μg/mL)
OLEE	130.02 ± 2.3	70.13 ± 1.2	100.00 ± 1.8
Trolox			15 µM ± 0.62

Values, expressed as mg gallic acid (GAE) and catechin (CE) equivalents, are the mean ± S.D. of three experiments in triplicate.

**Table 2 antioxidants-13-00073-t002:** Peaks list of selected secondary metabolites identified in OLEE.

Peak #	Rt, Min ^a^	Compound Tentative Identification	UV-Vis Data, nm ^b^	ESI- Data, *m*/*z* ^c^
1	6.1	hydroxytyrosol hexoside	280.2	315.0952 * (M-H); 153.0987
2	6.6	3,4 dihydroxyphenylacetic acid (DOPAC) ^d^	280.6	167.0419 (M-H); 190.0309 * (M-H+Na)
3	7.8	Hydroxytyrosol ^d^	275.9	153.0627 (M-H)
4	8.7	hydroxytyrosol derivative	279.1	315.0595 (M-H) (tr)
5	9.4	DOPAC derivative	280.4	315.0462; 181.0391 * (M-H)
6	13.0	eriodictyol hexoside-deoxyhexoside	283; 336	595.1698 * (M-H); 449.0234 (M-H-deoxyhex)
7	13.4	hydroxytyrosol derivative 2	280.1	315.0603 * (M-H)
8	13.8	naringenin hexoside-deoxyhexoside	284; 332sh	579.1677 * (M-H); 433.1583(M-H-deoxyhex)
9	14.5	apigenin hexoside derivative	266; 336	611.1039 * (M-H)
10	14.8	luteolin di-hexoside	257; 265; 346	609.1529 *(M-H); 447.1531 (M-H-hex)
11	15.1	luteolin hexoside-pentoside	253.4; 266.2; 345	579.1419 * (M-H)
12	15.8	methylapigenin di-hexoside	268; 337	607.1747 *(M-H); 445.1672 (M-H-hex)
13	15.9	ligstroside ^d^	288.2	523.1778 (M-H); 361.1532 (M-H-glc)
14	18.0	oleuropein aglycone	287	377.1309 (M-H)
15	18.6	eriodictyol deoxyhexoside	289; 335sh	433.1039 * (M-H)
16	19.3	naringenin deoxyhexoside	287; 332sh	417.1275 * (M-H)
17	20.8	luteolin 7-*O*-glucoside ^d^	248; 267;311	447.0097 * (M-H); 470.0989 (M-H+Na)
18	21.3	methylapigenin hexoside	272; 337	445.1209 * (M-H)
19	21.6	methylluteolin hexoside	249; 266; 345	461.1158 * (M-H)
20	23.3	luteolin deoxyhexoside	248; 269; 341	431.1049 * (M-H)
21	24.1	methylapigenin deoxyhexoside	274; 338	429.1258 * (M-H)
22	24.5	methylluteolin isomer 1	241; 263sh; 330	599.0658 * (2M-H)
23	25.2	methylapigenin isomer 1	270; 337	283.0679 * (M-H)
24	26.1	methylluteolin isomer 2	249; 269; 333	299.0629 (M-H); 599.0657 * (2M-H)
25	27.0	methylluteolin isomer 3	245; 269; 343	299.0621 * (M-H)
26	28.0	methylapigenin isomer 2	273; 337	590.1941 * (2M-H+Na); 283.0678 *
27	29.5	di-methylluteolin isomer 1	249; 268; 343	313.0773 (M-H); 627.1066 (2M-H)

^a^ as mean of three replicates; ^b^ from HPLC; ^c^ base peaks marked with an asterisk; ^d^ co-injection with pure commercial standard; sh shoulder.

## Data Availability

The data presented in this study are available on request from the authors.

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
