# Peer review of "Redox State Modulatory Activity and Cytotoxicity of Olea europaea L. (Oleaceae) Leaves Extract Enriched in Polyphenols Using Macroporous Resin"

_antioxidants, 2024, doi:10.3390/antiox13010073_

Round 1
Reviewer 1 Report
Comments and Suggestions for Authors
The manuscript antioxidants-2786331 is interesting and well written. However, there are a few shortcomings and deficiencies that need to be improved before publication.
My comments are included in the attached pdf file.

Author Response
R1: The manuscript antioxidants-2786331 is interesting and well-written. However, there are a few shortcomings and deficiencies that need to be improved before publication. My comments are included in the attached pdf file.
A: Dear Reviewer, thank you for your kind comments and suggestions to improve the quality of our work. We appreciate your efforts in reviewing the manuscript. We have studied the comments carefully and have made the expected corrections incorporating necessary revisions which we hope will meet your approval. Please see the attached pdf file for our point-by-point responses. The revised portions are highlighted in red in the revised version of the manuscript.

Reviewer 2 Report
Comments and Suggestions for Authors
Dear authors,
I read the article very carefully and I would like to formulate a series of observations and small recommendations.
In the introduction, you gave too much extension to the presentation of prostate cancer. In my opinion, lines 63-64 could be deleted. My recommendation is to reduce the introduction with a focus on the role of nutrition in the prevention of prostate cancer.
Why didn't the authors use reference substances for the performed tests (with the exception of trolox)?
In the results part, chapters 25.3. 2.6.1., 2.6.2, in my opinion there should be a sentence explaining why the authors chose those methods. I also think that their interpretation in the discussion part requires additional explanations, especially since the vegetable product is a very studied one.
In conclusion, my recommendation for this article is to publish it with minor revisions.
Author Response
R2: Dear authors,I read the article very carefully and I would like to formulate a series of observations and small recommendations.
A: Dear Reviewer, thank you for your kind comments and suggestions to improve the quality of our work. We appreciate your efforts in reviewing the manuscript. We have read the comments carefully and revised the manuscript accordingly. Below we have included the comments and our point-by-point responses. Revised portions are highlighted in red in the revised version of the manuscript.
Q: In the introduction, you gave too much extension to the presentation of prostate cancer. In my opinion, lines 63-64 could be deleted. My recommendation is to reduce the introduction with a focus on the role of nutrition in the prevention of prostate cancer.
A: The authors thank the reviewer for his suggestion, introduction was revised according to the reviewer's comments.
Q: Why didn't the authors use reference substances for the performed tests (with the exception of trolox)?
A: The authors thank the reviewer for her/his observation, but we would like to point out that our study was conducted on an enriched extract; for this reason, it is difficult to deduce the contribution of the single component to the activity of the phytocomplex and compare it to the effect of a reference substance in a cell model. However, we’ll take it into account to consider the introduction of a suitable reference substance for the next studies.
Q: In the results part, chapters 25.3. 2.6.1., 2.6.2, in my opinion there should be a sentence explaining why the authors chose those methods. I also think that their interpretation in the discussion part requires additional explanations, especially since the vegetable product is a very studied one.
A: The authors thank the reviewer for her/his suggestion, we revised the results part, chapters 3.3.2.; 3.4.1., 3.4.2., and the discussion part accordingly.
R2: In conclusion, my recommendation for this article is to publish it with minor revisions.
A: Thank you for your valuable comments.
Round 2
Reviewer 1 Report
Comments and Suggestions for Authors
The second version of manuscript antioxidants-2786331 is improved; currently, I have a few minor comments and after their correction I think the paper is ready to be published.
Comments are included in the attached pdf file.

Author Response
Dear Reviewer, thank you for your kind comments and suggestions to improve the quality of our work. We have studied the comments carefully and have made the expected corrections incorporating necessary revisions which we hope will meet your approval. Please see the attached pdf file for our point-by-point responses. The revised portions are highlighted in red in the revised version of the manuscript.
